# Ramucirumab and GSK1838705A Enhance the Inhibitory Effects of Low Concentration Sorafenib and Regorafenib Combination on HCC Cell Growth and Motility

**DOI:** 10.3390/cancers11060787

**Published:** 2019-06-07

**Authors:** Rosalba D’Alessandro, Maria Grazia Refolo, Palma Aurelia Iacovazzi, Pasqua Letizia Pesole, Caterina Messa, Brian Irving Carr

**Affiliations:** 1Laboratory of Cellular and Molecular Biology, Department of Clinical Pathology, National Institute of Gastroenterology, “Saverio de Bellis” Research Hospital, 70013 Castellana Grotte, BA, Italy; rosalba.dalessandro@irccsdebellis.it (R.D.); maria.refolo@irccsdebellis.it (M.G.R.); mina.iacovazzi@irccsdebellis.it (P.A.I.); pesoleletizia@gmail.com (P.L.P.); 2Department of Liver Cancer Biology, Liver Transplant Institute, Inonu University, Malatya 44280, Turkey

**Keywords:** combination therapy, synergism, multi-kinase inhibitors, vascular endothelial growth factor receptor, insulin-like growth factor 1 receptor, α fetoprotein, des-γ-carboxyprothrombin

## Abstract

Several new multikinase inhibitors have recently been introduced into clinical practice for hepatocellular carcinoma (HCC) therapy. Small increases in survival were reported as well as considerable toxicity. There is thus a need for effective therapies with lower toxicities. We examined whether a combination of sorafenib and regorafenib might also be effective at very low concentrations, with resulting potential for lessened clinical toxicity. MTT test, clonogenic assay, Ki67 staining and cell cycle analysis were assessed for cell proliferation and Annexin V and western blotting analysis relative to the expression of cleaved Caspase-3 and BID for cell apoptosis. In these experimental conditions cell growth and migration were potently inhibited and apoptosis induced even in HCC cells producing high alpha fetoprotein (AFP) levels (clinically worse prognosis). The combination also inhibited levels of the two HCC biomarkers, AFP and des gamma carboxy prothrombin (DCP). Additional inhibition of Vascular Endothelial Growth Factor Receptor (VEGFR) or Insulin-like Growth Factor 1 Receptor (IGF1R) enhanced effects on AFP and DCP levels, cell growth inhibition and MAPK and PI3K/Akt signaling inhibition due to sorafenib/regorafenib combination. These combinations have the potential for decreased toxicity while simultaneously enhancing therapeutic effects. This potential decrease in toxicity is being explored in ongoing studies.

## 1. Introduction

Until recently sorafenib was the only FDA approved systemic drug for the treatment of advanced hepatocellular carcinoma (HCC) [1]. Within the last few years, a number of other tyrosine kinase inhibitors (TKIs) have been investigated and lenvatinib represents the first breakthrough for first line therapy (REFLECT trial) [2] after multiple other trial failures and it is currently FDA approved as a treatment for patients with metastatic HCC. Improvements in patient outcomes have been demonstrated in randomized Phase III trials with regorafenib [3] and ramucirumab [4] as second line treatments after disease progression on sorafenib. The RESORCE trial showed that treatment with regorafenib, a molecule structurally related to sorafenib, provided a significant improvement in median overall survival in sorafenib-resistant patients, leading to a hypothesis that the inhibitory profiles of these drugs differ slightly [3]. Several growth factor receptors are inhibited by both these two TKIs including VEGFRs, c-Kit, PDGFR-b, whereas others, such asFGFR-1 and TIE-2 seem to be affected only by regorafenib [5,6]. Ramucirumab is an antagonistic anti-VEGFR2 monoclonal antibody extending overall survival duration in a subgroup of patients with high baseline serum α-fetoprotein (AFP) levels, as found in the phase III REACH-2 trial [4]. Several preclinical studies showed that other mitogens, like insulin-like growth factor (IGF), also have a crucial role in the growth and spread of HCC [7]. The IGF pathway provides an important mechanism modulating tumorigenesis and drug resistance in many tumors, including HCC [8]. However, although many inhibitors of IGF receptors (IGFR) have been developed and promising antitumor activity has been shown, clinical studies revealed very limited efficacy of IGFR inhibitors as single-agent in HCC therapy [9,10]. Despite the availability of these new targeted molecules, drug toxicity and resistance and subsequent tumor relapse are still difficult HCC management issues. Patients may develop resistance to a given treatment by activating alternative signaling that bypass the inhibitory effect of a single agent. Therefore, the choice of combining two or more drugs can be successful by inhibiting additional signaling pathways simultaneously and more effectively. In addition, combination therapy can offer the advantage of reducing drug doses and their toxicities, without compromising their effectiveness. The aim of the present study was to investigate the effects on cell growth and motility of the combined treatment of low doses of sorafenib and regorafenib. We also evaluated how the blockade of VEGFR2 with ramucirumab, and of IGF1R with GSK1838705A, enhanced synergistically the inhibitory action of the two multi-kinase inhibitors by acting on multiple signal pathways. The effects of ramucirumab and GSK1838705A were evaluated on two human HCC cell lines (PLC/PRF/5 and HepG2) characterized by different basal levels of AFP and DCP, in order to correlate the efficacy of specific drug treatments to the content of the two markers and possibly to detect changes in their expression following the treatments themselves.

## 2. Results

### 2.1. Inhibition of both VEGFR2 and IGF1R Potentiate the Effects on Cell Growth Deriving from the Combination of Regorafenib/Sorafenib in HCC Cell Lines

We initially examined whether addition of regorafenib could enhance sorafenib-mediated growth inhibition. Subsequently, all the data relative to sorafenib/regorafenib combination treatment were referred to by comparison with sorafenib single treatment as the reference point. A range of concentrations was examined for each of the drugs that were studied (as described in Materials and Methods section) and the lowest concentration of each drug that had a growth effect was subsequently used in all assays. PLC/PRF/5 and HepG2 human HCC cells were cultured in the presence of regorafenib and/or sorafenib administrated alone or in combination. Furthermore, the cells that were treated with regorafenib and/or sorafenib were also cultured with or without ramucirumab or GSK1838705A, as ramucirumab inhibits VEGFR2, while GSK1838705A inhibits IGF1-R. Cell proliferation was evaluated after 48 h of culture by MTT assay. Dose response curves relative to each single or combination treatment were performed in PLC/PRF/5 and HepG2 cells. The MTT optical density values were plotted in the graphs shown in Figure 1.

In PLC/PRF/5 cells the regorafenib and sorafenib IC_50_ values were 4.1 µM and 5.4 µM respectively. Treatment with the lowest concentration of regorafenib that had a growth-inhibitory effect (1 µM) in combination with increasing concentrations of sorafenib caused a further reduction of growth with an ensuing decrease of sorafenib IC_50_ value (4.4 µM vs. 5.4 µM). This inhibitory effect was enhanced when combining the lowest growth effective concentrations of regorafenib (1 µM) and sorafenib (2.5 µM) with increasing concentrations of both GSK1838705A and ramucirumab. In both treatment conditions the growth decrease resulted in a lowering of the IC50 values, 6.8 µM vs. 12.6 µM for GSK1838705A and 413 µM vs. 633 µM for ramucirumab. Comparable and more pronounced effects were also found in HepG2 cells which are known to be more sensitive to the action of the considered treatments. The growth decreases of effective single or combination treatments were expressed as percentage values and reported in Table 1.

The Combination Indexes (ICs) were computed for these drug combinations. Addition of a low concentration of regorafenib enhanced the inhibitory effect of low concentrations of sorafenib, being the CI for this combination below the line of additivity (CI = 0.9 in both cell lines). The addition of either ramucirumab or GSK1838705A further increased the growth inhibition exerted by regorafenib/sorafenib.

The CI values computed for the two cell lines ranged from 0.4 (PLC/PRF/5) and 0.2 (HepG2) for regorafenib/sorafenib/GSK1838705A and from 0.7 (PLC/PRF/5) and 0.5 (HepG2) for regorafenib/ sorafenib/ramucirumab combinations, respectively. These values were found to be well below the line of additivity (CI ≤ 1), showing that synergy was involved in these drug interactions (Figure 1). IGF1 and VEGF recombinant molecules were used as single treatments, in addition to the described treatments, in order to clarify the effects of IGF1R and VEGFR inhibitors. These experimental conditions were used to evaluate the ability of a single cell to grow into a colony in a clonogenic assay assessed for both HCC cell lines. The effects of single and combination drug treatments were determined after approximately two weeks. The results obtained for each drug treatment are reported in Figure 2A showing a representative assay performed in six-well plates. The number of colonies produced after each treatment was reported in the corresponding well (Figure 2A). The decrease in colony number of each single treatment with respect to control cells as well as the decrease of combination compared with single treatments was similar to the trend of those detected by MTT. These data were expressed as percentage values and reported in Appendix A. The effects of the described treatments on cell growth inhibition were evaluated by Ki67 staining, a well-known marker for cell proliferation, restricted to the phases G1, S, G2 and M of cell cycle. The results obtained for both cell lines indicated that after 24 h the combination of regorafenib and sorafenib was more effective than a single drug in reducing the fluorescence signal. The effect of GSK1838705A in enhancing the decrease of fluorescent signal exerted by the combination of regorafenib and sorafenib was significant in PLC/PRF/5 but absent in HepG2 cells in which the stimulatory action of IGF1 was also not relevant. By contrast, the addition of ramucirumab to regorafenib and sorafenib combination caused a significant lowering in Ki67 signal in both cell lines, although VEGF stimulatory action was more pronounced in HepG2 cells (Figure 2B and Appendix A). To further confirm the inhibitory effects of the considered drug treatments, the progression on cell cycle was also assessed. The cells were synchronized in the S phase of the cell cycle (T0) with 0.2 M thymidine and after 3 h from block release (T1) the percentage of cells in G0/G1, S and G2/M phases was evaluated by the Cell Cycle Kit. Although it was observed only a small decrease in G2/M transition after the combined regorafenib and sorafenib treatment compared to single treatments, the addition of GSK1838705A and especially of ramucirumab determined a further and significant inhibition in the cell cycle progression in both cell lines (Figure 2C and Appendix A).

### 2.2. Inhibition of both VEGFR2 and IGF1R Potentiate the Effects on Cell Apoptosis Deriving from the Regorafenib/Sorafenib Combination

Using the same experimental conditions, we also analyzed apoptosis (Figure 3 and Appendix A) which, along with proliferation, determines the state of cell growth.

In PLC/PRF/5 cells, regorafenib added simultaneously to sorafenib caused an increase of 22.9% in cellular Annexin V compared with sorafenib-only treated cells. The addition of GSK1838705A caused a further increase of 27.8% compared to the combination of regorafenib and sorafenib. If Ramucirumab was added at the regorafenib/sorafenib combination, a stronger effect on apoptosis was observed (68.6%). A similar trend was found in HepG2 cells with the difference that the combined effect of regorafenib and sorafenib is more pronounced in this cell line (31.6% respect to sorafenib treated cells). Moreover, ramucirumab, which alone had a significant effect in inducing apoptosis (76.8% respect to control cells), was able to further enhance this process in combination with regorafenib and sorafenib (38.4% more than the double treatment) (Figure 3A). Western blotting experiments were also performed to investigate the activation status of Caspase-3 and BID, two pro-apoptotic markers. Activated cleaved Caspase-3 and BID were expressed at comparable levels in single or combination regorafenib and sorafenib-treated cells, whereas cleaved Caspase-3 was significantly activated after the addition of GSK1838705A and ramucirumab to regorafenib/sorafenib double treatment in both cell lines (Figure 3B).

### 2.3. Inhibition of both VEGFR2 and IGF1R Potentiate the Effects on Cell Migration Deriving from the Regorafenib/Sorafenib Combination

To test the effects of either GSK1838705A or ramucirumab on regorafenib/sorafenib-mediated inhibition of cell migration, PLC/PRF/5 and HepG2 cells were seeded onto Oris plates, coated with collagen I and fibronectin matrix, and then the cells were treated with drugs according to the experimental conditions described. Microscopic analysis was assessed both immediately after stoppers removal (T0) and at later times. The percentage of migration after 48 h was reported in the graphs represented in Figure 4A and Appendix A.

In PLC/PRF/5 cells the effect of combined treatment of regorafenib and sorafenib was potentiated of 31% respect to the sorafenib single treatments, the actions of both GSK1838705A and especially of ramucirumab were more pronounced, the first inhibiting the percentage of migration of 40% the second of 73.5% with respect to the regorafenib/sorafenib combination treatment. Although in the HepG2 cells the inhibitory effect of the regorafenib/sorafenib combination was comparable to single agent treatments, the effect of GSK1838705A was moderate as single treatment (39% of inhibition respect to untreated cells) and was able to increase the inhibitory effect of regorafenib/sorafenib (47.5% of inhibition more than the regorafenib/sorafenib double treatment). Cell migration was almost completely blocked by ramucirumab single treatment (95% of inhibition respect to untreated cells). The effects were the same independently of the matrix used. The analysis was extended to the action of these drug combination on the organization of cytoskeleton. The staining with DyLight 554 Phalloidin performed after 24 h of single drug treatment revealed a strong reduction and depolymerization of actin fibers (F-actin). Combination treatments resulted in an almost complete loss of F-actin fibers in the cytoplasm (Figure 4B and Appendix A).

### 2.4. Inhibition of both VEGFR2 and IGF1R Potentiate the Reduction of AFP and DCP Secretion Due to the Regorafenib/Sorafenib Combination

All the analyzes presented were performed in two HCC cell lines characterized by a different secretion of AFP and DCP, two markers associated with HCC cell growth and motility. PLC/PRF/5 cells are characterized by low levels of AFP secretion and high DCP, whereas HepG2 cells by contrast secrete high levels of AFP [11] and low of DCP (see Materials and Methods section). The secreted levels of AFP and DCP in the cell culture medium were measured after 48 h of treatments described. Figure 5A and Appendix A showed that, in both cell lines, the inhibitory effect on AFP secretion caused by treatment with regorafenib/sorafenib combination (27% for PLC/PRF/5 and 17% for HepG2 in comparison to sorafenib-alone treated cells) was increased both by the addition of GSK1838705A and by addition of ramucirumab, with a substantial difference in the two cell lines. In PLC/PRF/5 cells the most evident effect was given by the addition of GSK1838705A (45% more than the regorafenib/sorafenib combination treatment) while ramucirumab caused only a minor decrease (7.6% more than the combination treatment).

In HepG2 cells the effect of ramucirumab addition was stronger (37.9% more than the regorafenib/sorafenib combination treatment) than that evoked by GSK1838705A (15% more than the double treatment). The inhibition of DCP secretion induced by sorafenib/regorafenib combination treatment was greater in PLC/PRF/5 cells (68.5% more than sorafenib single treatment) than in HepG2 cells (20% more than sorafenib single treatment). Moreover, GSK1838705Awas able to potentiate the inhibition of the combination treatment by a further 37% only in PLC/PRF/5 cells. By contrast, ramucirumab exerted major effect on HepG2 cells increasing the percentage of double treatment inhibition of 25% (Figure 5B and Appendix A).

### 2.5. Inhibition of either VEGFR2 or IGF1R Potentiates the Effects on MAPK and PI3K/Akt Signaling Due to the Regorafenib/Sorafenib Combination

Cells treated with regorafenib and/or sorafenib were additionally cultured with or without GSK1838705A or ramucirumab. IGF1 caused an increase in the expression of its receptor, and also a stimulation of the expression of VEGFR in particular in PLC/PRF/5 cells. Both regorafenib and sorafenib administrated either alone or in combination did not modify igf1r expression, but the simultaneous addition of GSK1838705A resulted in a substantial reduction. VEGF did not modify the expression of either its receptor nor of IGF1R. Multikinase inhibitors showed only minor inhibitory effect on the expression of VEGFR both after single and combination treatments, while the addition of ramucirumab was able to strongly inhibit their expression, especially in HepG2 cells (Figure 6A).

Some of the key molecules involved in the signal transduction of both IGF1R and VEGFR activated cascades were analyzed by western blotting and Muse Activation Dual Detection Kits. MAPK signaling activation was evaluated by measuring the extent of ERK phosphorylation relative to the total ERK expression by western blotting (Figure 6B) and Muse MAPK Activation Dual Detection Kit (Figure 6C). In PLC/PRF/5 cells the combined treatment of regorafenib/sorafenib showed a major inhibition of ERK phosphorylation (14%) compared to sorafenib administered as a single treatment. Addition of GSK1838705A to combination regorafenib/sorafenib treatment caused an additional decrease of 8% in pERK levels. A greater effect was caused by the addition of ramucirumab, which caused a further decrease in ERK phosphorylation levels equal to 21%. In HepG2 cells, GSK1838705A and ramucirumab increased almost similarly the inhibitory effect of the combination of regorafenib/sorafenib on ERK phosphorylation, in particular the 9% inhibition caused by the combination treatment when compared to sorafenib single treatment was increased by a further 17% in the case of addition of GSK1838705A to the culture medium and 18% in the case of addition of ramucirumab (Figure 6C and Appendix A). Activation of PI3K/Akt pathway in the same experimental conditions was also investigated, using western blotting and confirmed and quantified by the Muse PI3K Activation Dual Detection Kit. In PLC/PRF/5 cells, regorafenib/sorafenib combination caused a decrease in percentage of activated p-Akt (16.9% lower than the single treatment with sorafenib). This decrease reached 18.8% following the addition of the GSK1838705A to the combination treatment and 24.6% when ramucirumab was added with regorafenib/sorafenib combination. In HepG2 cells the effect of combination regorafenib/sorafenib on Akt phosphorylation levels was almost comparable with the results found in the PLC/PRF/5 cells, in fact the combined treatment caused a 11% reduction in phosphorylation compared to treatment with sorafenib alone. In this cell line the effects evoked by the simultaneous addition to regorafenib/sorafenib treatment of GSK1838705A and especially of ramucirumab were much more evident than in the PLC/PRF/5 cells. In particular, the percentage of p-Akt activation decreased by a further 41% with the addition of GSK1838705A and 51% with ramucirumab (Figure 6C and Appendix A).

## 3. Discussion

Increasingly deeper knowledge of the biology of HCC has permitted significant progress in HCC therapy in recent years. Currently, sorafenib is no longer the only small molecule drug used in the first line of therapy for HCC, as the FDA has recently also approved lenvatinib, another tyrosine kinase inhibitor [1,2]. Nevertheless, the expected survival of these patients is only a few months following first-line treatment if they don’t go on to second-line therapy. Regorafenib was the first drug to show a survival benefit in patients who had HCC progression on sorafenib [3]. Regorafenib extends the kinase spectrum inhibited by its analogue sorafenib having greater potency against angiogenic and stromal receptors VEGFR-1-3, TIE-2, FGFR1 and PDGF-βR as well as the oncogenic receptors KIT and RET [5,6,12,13,14]. However, a worsening of quality of life (QoL) outcomes was highlighted. Recently, REACH-2 trial reported that ramucirumab treatment was correlated with maintenance of QoL and with a superior median overall survival duration in a subgroup of patients with a baseline serum AFP ≥400 ng/mL [4]. The impact of this study has been to identify for the first time a biomarker-selected population of patients (AFP) with a poor prognosis [15,16,17]. Moreover, our previous in vitro studies suggested that resistance to multikinase inhibitors or chemotherapeutic agents may be due to several milieu factors [18,19,20]. Shared escape mechanisms for both sorafenib and regorafenib have been speculated in HCC cells treated with IGF1 suggesting a key role for this factor in antagonizing drug-mediated growth, migration and invasion inhibition, as well as the drug-mediated induction of apoptosis. It has been demonstrated that the IGF1 treatment was correlated with an over expression of its major down-stream pathway PI3K-Akt [21]. Moreover a strong contribution of GSK1838705A in enhancing both sorafenib and regorafenib effects has been highlighted [18,20], suggesting a close relation between the IGF1-R and PI3K/Akt/mTOR signaling pathways and HCC biology. Hence, simultaneous inhibition of multiple signaling cascades could represent a crucial approach for HCC management, since the synergistic effects of treatments with combinations of drugs offer the opportunity to reduce the drug doses without affecting their effectiveness. Based on the results from preclinical studies, in which only partial overlap of the mechanisms of sorafenib and regorafenib was found, and clinical studies, in which the two inhibitors were used sequentially, in the present work the effects on cell growth and motility of a combined treatment of low doses of the two kinase inhibitors was evaluated. Based on three different proliferation assays in which drug effects were evaluated at different times, our results clearly showed that regorafenib enhanced the inhibitory effect of sorafenib on cell proliferation. Both drugs have been used at ten times lower concentrations than their IC_50_ and although the effects of single treatment were minimal, the combined effect can be considered slight synergistic since CIs computed for these drug combinations were found to be just below the line of additivity (CI ≤ 1) for both cell lines. Even more significantly synergistic effects were observed if GSK1838705A or ramucirumab were added to the combined treatment of sorafenib/regorafenib. The inhibitory effect of GSK1838705A in a single treatment can be considered mild, but the drug synergy was moderate (CI = 0.4) for the triple combination sorafenib/regorafenib/GSK1838705A in PLC/PRF/5 cells and strong (CI = 0.2) for the same combination in HepG2 cells. With respect to IGF1R inhibitor, ramucirumab, already having a stronger inhibitory effect in single treatments, had a less strong but significant synergistic effect when added to the sorafenib/regorafenib combination (CI = 0.7 in PLC/PRF/5 and CI = 0.5 in HepG2). Three different proliferation experiments, clonogenic assay, Ki67 staining and cell cycle progression were performed to confirm the potentiation of the inhibitory effect obtained with combined treatments in both cell lines investigated. Moreover, the addition of ramucirumab to sorafenib/regorafenib combination resulted always more efficacious in HepG2 cells. Cell growth is a result of the balance between proliferation and apoptosis. The analysis of apoptosis, by the evaluation of Annexin V on the apoptotic cell membrane and the expression of two pro-apoptotic markers such as cleaved Caspase-3 and BID, confirmed both an increase in apoptosis in the combined treatments of sorafenib/regorafenib compared to single treatments and a further induction of apoptosis in the combination treatments to which either GSK1838705A or ramucirumab were added. The addition of ramucirumab was particularly effective, resulting in a 77% increase in apoptotic cells compared to the percentage observed in sorafenib/regorafenib-treated HepG2 cells. All the drugs investigated exerted their antitumor activity also in inhibiting cell motility [7,18,20,21]. Inhibition of cell motility was evident already after single treatments, therefore the combination of sorafenib/regorafenib did not significantly enhance this effect. By contrast, GSK1838705A or ramucirumab exerted even more pronounced effects in enhancing the combined action of sorafenib/regorafenib. In particular, the triple combination with ramucirumab was able to block cell migration almost completely. Here, we provide evidence that the combination treatment with sorafenib/regorafenib as well as the further addition of IGF1-R or VEGFR antagonists caused a significant reduction and depolymerization of F-actin, resulting in synergistic inhibition of cell migration. It is widely accepted that AFP and in particular DCP are two molecules related to HCC cell motility and aggressiveness [22,23,24]. AFP is a glycoprotein whose synthesis is normally repressed in adulthood. The serum AFP concentration of 20 ng/mL represents a value to differentiate patients with and without HCC. High serum AFP concentration (>400 ng/mL) is associated with greater tumor size, portal vein thrombosis and poor prognosis [24]. Results of REACH-2 trial pointed to elevated AFP levels at diagnosis as a biomarker to select patients who benefit from ramucirumab therapy [4]. Although the measurement of AFP is useful in the screening HCC patients, it is not able to differentiate HCC from benign hepatic disorders. Therefore, further efforts are needed to identify other HCC biomarkers. Des-γ-carboxy- prothrombin (DCP) is a protein that is found increased in the serum of HCC patients. Several studies have indicated that the levels of AFP and DCP do not correlate. However, when used together, the DCP and AFP measurements enhance the sensitivity of identifying HCC patients [22,23]. Our previous finding revealed that in HCC both sorafenib and regorafenib inhibited AFP secretion in HCC cells, while platelet containing growth factors caused an increase in AFP release [25,26]. In addition, GSK1838705A has been shown to be ineffective in modifying the levels of secretion of AFP both when given alone or in combination with the regorafenib [20]. Previous results have been confirmed in the present study and new and interesting elements have been added. In PLC/PRF/5 cells that have low AFP and high DCP content, major effects were observed with the combined treatment sorafenib/regorafenib that caused a significant decrease in AFP release and the addition of GSK1838705A to this combination further inhibited AFP secretion. The inhibitory effects of the combination sorafenib/regorafenib and of the triple combination with the GSK1838705A were even more evident for DCP protein secretion. In HepG2 cells, even if a major inhibitory effect was detectable following the combined sorafenib/regorafenib treatment with respect to single agent treatments, addition of ramucirumab to the combination treatment led to a substantial reduction especially of AFP release but also of DCP. Combination treatments permitted inhibition of multiple signal pathways simultaneously making compensating mechanisms (and thus drug resistance) more difficult [16,18,20]. The present study highlighted the role of IGF1 and VEGF in stimulating growth and motility by the activation of their receptors leading to the activation of crucial kinases involved in the signal transduction. IGF1 caused not only an increase in the expression of its receptor, but was also able to stimulate the expression of VEGFR, especially in PLC/PRF/5 cells. These results support the hypothesis that possible cross-talk can occur among different receptors. Therefore, the simultaneous inhibition of both receptors can overcome the reciprocal compensation between the signaling cascades. Our data indicated that the expression of both IGF1R and VEGFR reflected strongly the growth inhibition in cells treated with a combination of multikinase inhibitors with GSK1838705A or ramucirumab. Furthermore, combination treatments were shown to be particularly effective in the inhibition of some of the key molecules involved in the signal transduction of both IGF1R and VEGFR activated cascades. The activation status of MAPK and PI3K/Akt signaling, two of the main targets of the drugs considered, was investigated [18,20,21]. Both sorafenib and regorafenib are known to exert their major effect through the inhibition of MAPK signaling, whereas both GSK1838705A and ramucirumab affect both molecular cascades as well as PI3K/Akt [7,27,28]. Our analyses confirmed that both tyrosine kinase inhibitors used at low concentration (lower than their respective IC50) affected ERK phosphorylation but they were almost ineffective on Akt phosphorylation [18,20]. However, their effects were both potentiated by their combination. Moreover, our results revealed that both GSK1838705A and ramucirumab were able to potentiate the sorafenib/regorafenib combination in inhibiting both MAPK and PI3K/Akt signaling, in particular their action was stronger on PI3K/Akt. Interestingly, ramucirumab effects, also as single treatment, were more evident in high AFP HepG2 cells. The VEGFs through the activation of VEGFRs on endothelial cells regulate angiogenesis, one of the main process involved in tumor growth and spreading. Moreover, autocrine VEGF signaling in several tumor type, occurs when VEGF secreted by tumor cells binds VEGFRs present on the tumor cell membrane, thus contributing to stimulation of cell growth. Recent data confirmed the presence of this autocrine signal also in HCC cells [29]. All the presented data pointed for the first time the attention on the modulatory effect of regorafenib added to sorafenib in combination treatments, showing that the two multikinase inhibitors that until now have been used in clinical practice sequentially could potentially give good HCC control results in combination treatments. This synergistic effect could be potentially improved if other specific inhibitors, such as GSK1838705A or ramucirumab, were added to sorafenib/regorafenib combination to obtain major effects also on other signal pathways. The selection of these agents proves to be strategic and the analysis of bio markers such as AFP or DCP or both can provide a useful resource of guiding the choice.

## 4. Materials and Methods

### 4.1. Cells and Drugs

PLC/PRF/5 and HepG2 human HCC cell lines were purchased from the National Institute of Biomedical Innovation JCRB Cell Bank (Osaka, Japan) and cultured with Dulbecco’s Modified Eagle’s Medium (DMEM). All cell culture supplements were purchased from Sigma-Aldrich (Milan, Italy).

Regorafenib and sorafenib were a gift from Bayer Corp (West Haven, CT, USA); ramucirumab was purchased from Eli Lilly (Utrecht, The Netherlands), GSK1838705A was purchased from Selleck Chemicals (Houston, TX, USA). Recombinant Insulin-like Growth Factor 1 (IGF1) was purchased from Calbiochem (San Diego, CA, USA) and recombinant Vascular Endothelial Growth Factor from ORF Genetics (Kopavogur, Iceland). The two cell lines were grown in monolayer culture with 10% fetal bovine serum (FBS), 100 U/mL penicillin and 100 μg/mL streptomycin addition and incubated at 37 °C in a humidified atmosphere containing 5% CO_2_ in air.

### 4.2. Cell Proliferation and Drug Synergy Evaluation

#### 4.2.1. MTT Assay

PLC/PRF/5 and HepG2 cells were cultured in medium containing different concentrations of regorafenib (1, 2.5, 5 Μm for PLC/PRF/5 and 0.1, 0.5, 1 μM for HepG2), sorafenib (1, 2.5, 5 μM), GSK1838705A (1, 2, 3 μM) and ramucirumab (200, 400, 600 μM) used alone and in combination as indicated in the Results section. After 48 h of incubation, the vital and proliferating cells ware estimated by a colorimetric 3-(4,5 dimethylthiazol-2-yl)-2,5-diphenyltetrazolium bromide (MTT) test. The results were representative of three independent experiments. For each drug treatment, the OD values derived from the MTT assay were plotted in dose-response curves were and relative IC_50_s were calculated using Microsoft Office Excel based on the linear curve equation (y = mx + c). The potential synergistic, additive or antagonistic effects of drugs were assessed experimentally and data derived from MTT assay were implemented in CompuSyn software (Biosoft, Cambridge, UK) based on the method described by Chou and Talalay [30,31]. The Chou and Talalay approach consider drug potency as well as the relationship between dose and response for each drug. Results are reported as the Combination Index (CI). Values of CI < 1, CI ± 1, and CI > 1 imply synergy, additivity, and antagonism, respectively. In all the subsequent combined treatments each drug was used at the lowest concentration showing CI < 1, in particular cells were cultured with 1 μM (PLC/PRF/5) or 0.1 μM (HepG2) regorafenib, 2.5 μM (PLC/PRF/5) or 1 μM (HepG2) sorafenib, 3 μM GSK1838705A and 400 μM ramucirumab administrated singularly or in combination.

#### 4.2.2. Clonogenic Assay

PLC/PRF/5 and HepG2 cells were harvested from a stock culture and plated at appropriate dilutions (500 cells/well for PLC/PRF/5 and 250 cells/well for HepG2 cells) into six-well dishes. After attachment of the cells to the dishes, treatments were performed before cells start replicating; IGF1 75 ng/mL and VEGF 20 ng/mL recombinant molecules were used as single treatment, in addition to the described treatments. Then the dishes were placed in an incubator and left there for a time equivalent to at least six potential cell divisions (clones are considered to represent viable cells if they contain in excess of 50 cells).Fixation and staining protocol provided the use of absolute methanol for 10 min at room temperature and Crystal Violet 0.5% in 20% methanol for 15 min. Plates with colonies were left to dry in normal air at room temperature. Photographs were taken of each six-well plate and analyzed for the count by ImageJ software (http://rsb.info.nih.gov/ij/). The Plating Efficiency (PE, the fraction of colonies from cells not exposed to the treatment) of control cells was determined and the surviving fraction of cells after any treatment was always calculated taking into account the PE of control cells. Data were expressed as percentage decrease in colony formation, considering 100% the colonies formed from untreated control cells. The results were representative of three independent experiments.

#### 4.2.3. Ki67 Staining

HCC cell lines were cultured for 24 h in 1% FBS medium and seeded in 96 multi-well plates. After attachment of the cells to the dishes, in addition to the described treatments, IGF1 75 ng/mL and VEGF 20 ng/mL recombinant molecules were also used as single treatment. The proliferating cells were evaluated by the staining with anti-human Ki-67 (BioLegend Inc., San Diego, CA, USA) diluted 1:200 in PBS (Cell Signaling, Beverly, MA, USA) for 2h in a humidified dark chamber. The staining was performed as previously described [32]. The ImageJ software (http://rsb.info.nih.gov/ij/) was used for the analysis of fluorescent signals and the values relative to three independent experiments were plotted in graphs using GraphPad Prism 5.0 software (La Jolla, CA, USA).

#### 4.2.4. Cell Cycle Analysis

PLC/PRF/5 and HepG2 cells were synchronized adding 0.2 M thymidine to the medium, as previously described [32]. Cells were than separated into two groups: the first group was collected for the analysis at T0 and the second one continued culturing as described for clonogenic assay. After 3 h HCC cells were collected and processed by Muse Cell Cycle Kit (Millipore, Darmstadt, Germany) as previously described [32]. The values representative of three independent experiments were plotted in the relative graphs created with GraphPad Prism 5.0 software.

### 4.3. Apoptosis

PLC/PRF/5 and HepG2 cells were cultured as described above. Flow cytometry technology (Muse Cell Analyzer, Millipore, Darmstadt, Germany) was employed for quantitative analysis of live, early/late apoptotic and dead cells based on the activation status of Annexin V. Cell marker (7-AAD) was also used for dead cells. The cells were then processed following the user’s guide. Three independent experiments were performed in triplicate and the results were indicated in the graphs as means ±SD. Activated Caspase-3 and BID markers were evaluated in PLC/PRF/5 and HepG2 cells treated for 48h as indicated above with the following antibodies: Cleaved Caspase-3 (Asp175), BID and β-Actin (13E5) (Cell Signaling). Western blot analysis was performed as previously described [25].

### 4.4. Migration Assays

PLC/PRF/5 and HepG2 cells were seeded onto Orris plates (Platypus Technologies, Madison, WI, USA) that provided wells coated with collagen I or fibronectin. Briefly, after the adhesion, cells were subjected to above described treatments. When the stoppers that delimit the detection zone have been removed the cells started to migrate. Cells were examined microscopically and photographs of each well were taken after the stoppers removal (T0) and after 24 h (T1), 48 h (T2) and 72 h (T3). The values were expressed as percentage of migration, with 100% representative of detection zone completely closed. Three independent experiments were performed in triplicate. Moreover, the organization of cytoskeleton of PLC/PRF/5 and HepG2 cells were evaluated by the staining with DyLight 554 Phalloidin (Cell Signaling). Cells were cultured for 24 h in 1% FBS medium and then were trypsinized, resuspended in 1% FBS medium and seeded in 96 multi-well plates. After attachment of the cells to the dishes, in addition to the described treatments, IGF1 75 ng/mL and VEGF 20 ng/mL recombinant molecules were also used as single treatment. According to the immunostaining protocol above described, HCC cells were immunolabeled in humidified dark chamber with DyLight 554 Phalloidin for 2 h and with DAPI for 10 min to mark nuclei. A ZOE fluorescent cell imager (Bio-Rad, Milan, Italy) was used to acquire images.

### 4.5. AFP and DCP Measurement

PLC/PRF/5 and HepG2 cells were cultured according to the above described treatment. UniCel Integrated Workstations DxC660i (Beckman–Coulter, Fullerton, CA, USA), based on a chemiluminescent immunometric method, was used as automated system to measure AFP and DCP levels in culture media. The basal levels of AFP were found <20 ng/mL in PLC/PRF/5 and >20 ng/mL in HepG2 cells. The basal level of DCP were found >40 mU/mL in PLC/PRF/5 and <40 mU/mL in HepG2 cells. Results were representative of three independent experiments.

### 4.6. MAPK Activation

PLC/PRF/5 and HepG2 cell lines were cultured in presence of regorafenib, sorafenib, GSK1838705A and ramucirumab alone or in combination for 48 h. According to the western blotting protocol above described, the levels of the protein investigated were evaluated with the following antibodies: of Phospho-p44/42 MAPK(Erk1/2) (Thr202/Tyr204) and p44/42 MAPK(Erk1/2), β-Actin (13E5) (Cell Signaling). Muse MAPK Activation Dual Detection Kit (Millipore), was used to measure total levels of ERK. This kit included a phospho-specific antiphospho-Erk1/2 (Thr202/Tyr204, Thr185/Tyr187)-Phycoerythrin and an anti-Erk1/2-PECy5 conjugated antibody, to measure the amount of MAPK phosphorylation relative to the total MAPK expression in the cells processed according to the user’s guide. The levels of both the total and phosphorylated protein can be evaluated in the cell in the same time, resulting in a normalized measurement of MAPK activation after treatments. PLC/PRF/5 and HepG2 cell lines were cultured in presence of Regorafenib, Sorafenib, GSK1838705A and Ramucirumab alone or in combination for 15 minas it was done in the experiments previously described. Results were representative of three independent experiments.

### 4.7. PI3K Activation

PLC/PRF/5 and HepG2 cell lines were cultured in presence of regorafenib, sorafenib, GSK1838705A and ramucirumab alone or in combination for 48h. According to the western blotting protocol above described, the levels of the protein investigated were evaluated with the following antibodies: Phospho-PI3K p85(Tyr458)/p55(Tyr199) and PI3Kinase p110α, Phospho-Akt (Ser473) and Akt (pan) (11E7), β-Actin (13E5) (Cell Signaling). Muse PI3K Activation Dual Detection Kit (Millipore), was used to observe the fluorescent signal emitted by a specific anti-phospho-Akt (Ser473), Alexa Fluor 555, and an Akt-PECy5 conjugated antibody. This two-color kit is useful to measure the extent of Akt phosphorylation relative to the total Akt expression. The levels of both the total and phosphorylated protein can be evaluated in the cell in the same time, resulting in a normalized measurement of Akt activation after treatments. PLC/PRF/5 and HepG2 cells were treated in presence of the indicated concentrations of regorafenib, sorafenib, GSK1838705A and ramucirumab alone or in combination for 24 h respecting the drug concentrations indicated above. Results were representative of three independent experiments.

### 4.8. Statistical Analysis

GraphPad Prism 5.0 software was used for all statistical analysis. The differences between two unmatched groups were evaluated by Mann–Whitney non parametric test. *p* < 0.05 was considered statistically significant. All experiments were done in triplicate and data are presented as mean ± standard deviation (SD).

## 5. Conclusions

Given the high toxicity in patients of both Sorafenib and Regorafenib, their combination using lower doses or the multiple combinations with other agents, that synergistically enhance their effectiveness, are a source of interest in clinical practice in offering the possibility of lowering their toxicity while simultaneously enhancing their effectiveness. This aspect is currently being investigated.

## Figures and Tables

**Figure 1 cancers-11-00787-f001:**
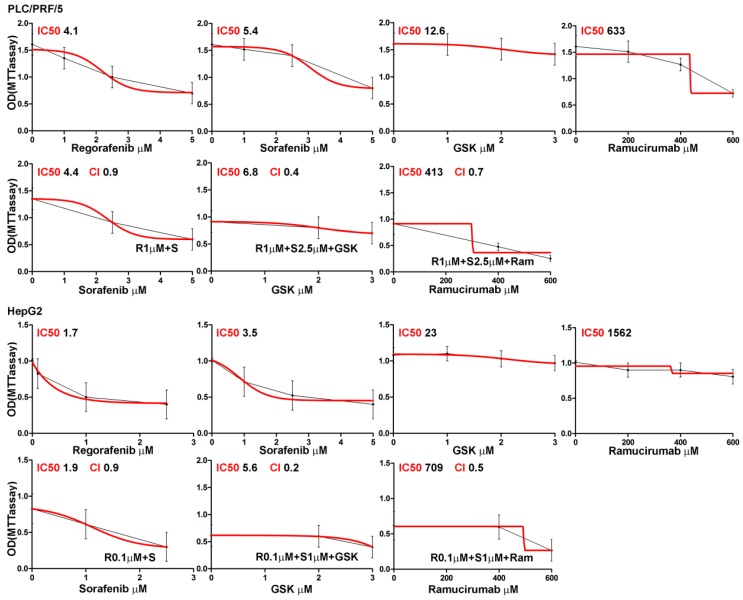
Dose response curves relative to each single or combined treatment in PLC/PRF/5, and HepG2 cells. PLC/PRF/5 and HepG2 cells were cultured in medium containing different concentrations of regorafenib (1, 2.5, 5 μM for PLC/PRF/5 and 0.1, 0.5, 1 μM for HepG2), sorafenib (1, 2.5, 5 μM), GSK1838705A (1, 2, 3 μM) and ramucirumab (200, 400, 600 μM) used alone and in combination for 48 h. Dose-response curves were generated for each drug or drug combination and relative IC50 and the CIs were indicated on the respective graph. The results of three independent experiments are expressed as means ± SD.

**Figure 2 cancers-11-00787-f002:**
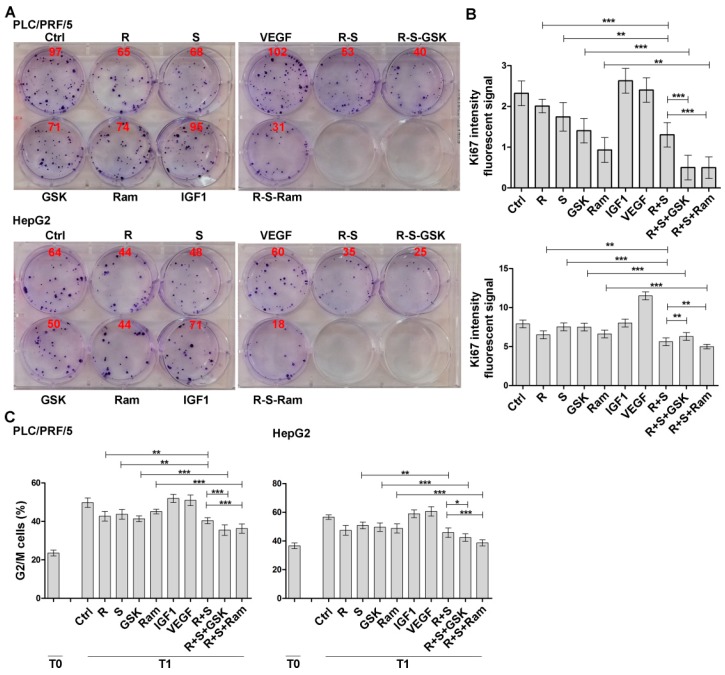
GSK1838705A and ramucirumab potentiate the anti-proliferative effects of sorafenib/regorafenib combination in HCC cells. Cells were cultured with 2.5 µM (PLC/PRF/5) or 1 µM (HepG2) sorafenib, 1 µM (PLC/PRF/5) or 0.1 µM (HepG2) regorafenib, 3 µM GSK1838705A and 400 µM ramucirumab administrated singularly or in combination. IGF1 75 ng/mL and VEGF 20 ng/mL recombinant molecules were used as single treatment. (**A**) Clonogenic assay was performed after 2 weeks. The results obtained for each drug treatment are reported showing a representative assay performed in six-well plates. The number of colonies produced after each treatment is reported in the corresponding well. (**B**) For each experimental condition, the fluorescent signal intensity was measured and values, expressed as mean ± SD, plotted in the relative graph. ** *p* < 0.001; *** *p* < 0.0001. (**C**) Cells were synchronized in the S phase of the cell cycle (T0), after 3 h from block release (T1) the percentage of cells in G2/M phases was evaluated and plotted in the graphs. The results of three independent experiments, expressed as mean ± SD, are plotted in the relative graphs. * *p* < 0.05; ** *p* < 0.001; *** *p* < 0.0001.

**Figure 3 cancers-11-00787-f003:**
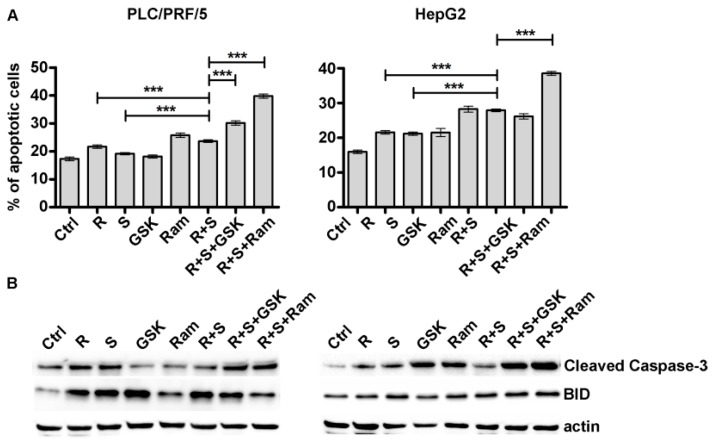
GSK1838705A and ramucirumab potentiate the inhibitory effects of Sorafenib/Regorafenib combination on HCC cell apoptosis. PLC/PRF/5 and HepG2 cells treated respectively with 2.5 µM and 1 µM sorafenib, 1 µM and 0.1 µM regorafenib, 3 µM GSK1838705A and 400 µM ramucirumab administrated as single or combined treatments. (**A**) Muse Annexin V Cell Assay was assessed after 48h. Three independent experiments were performed and the results are expressed as means ± SD. *** *p* < 0.0001. (**B**) Western blotting showing the expression levels of cleaved Caspase-3 and BID after 48 h single or combined treatments.

**Figure 4 cancers-11-00787-f004:**
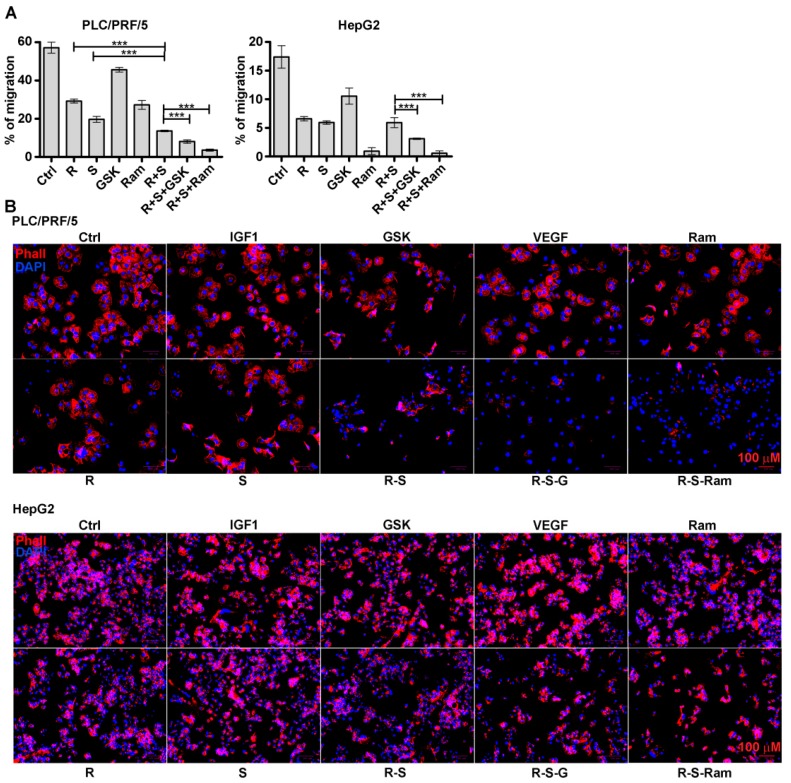
GSK1838705A and ramucirumab potentiate the inhibitory effects of sorafenib/regorafenib combination on HCC cell motility and depolymerization of actin cytoskeleton. Cells were cultured with 2.5 µM (PLC/PRF/5) or 1 µM (HepG2) sorafenib, 1 µM (PLC/PRF/5) or 0.1 µM (HepG2) regorafenib, 3 µM GSK1838705A and 400 µM ramucirumab administrated singularly or in combination. (**A**) Migration assay in PLC/PRF/5 and HepG2 cultured on fibronectin coated wells. The percentage of migration were calculated at the time T0 and after 48 h (T2). The 100% represents the detection zone completely closed. The experiments were performed in triplicate and the mean values ± SD are plotted in the relative graph. *** *p* < 0.0001. (**B**) In addition to the described treatments, IGF1 75 ng/mL and VEGF 20 ng/mL recombinant molecules were used as single treatment. Representative cell staining with DyLight 554 Phalloidin. Scale bar: 100 µm.

**Figure 5 cancers-11-00787-f005:**
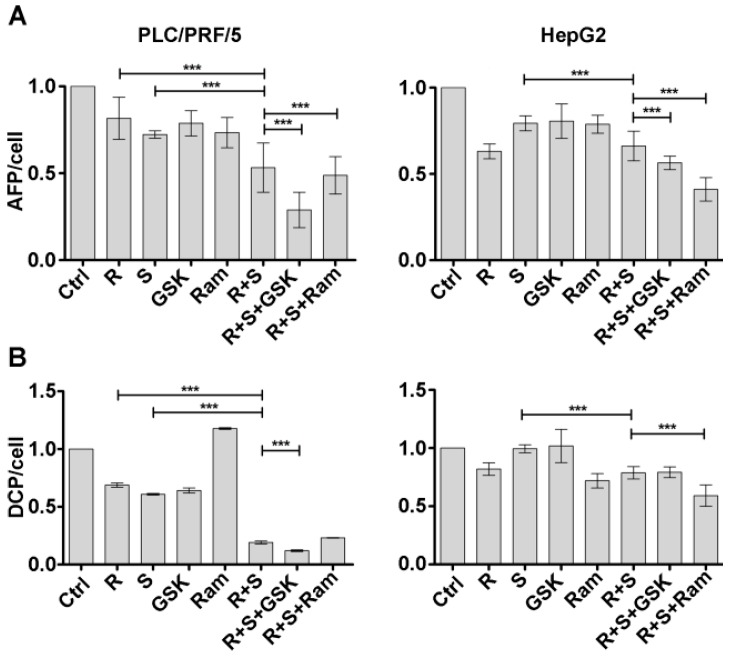
GSK1838705A and ramucirumab potentiate the inhibitory effects of sorafenib/regorafenib combination on AFP and DCP secretion in HCC cells. Cells were cultured with 2.5 µM (PLC/PRF/5) or 1 µM (HepG2) sorafenib, 1 µM (PLC/PRF/5) or 0.1 µM (HepG2) regorafenib, 3 µM GSK1838705A and 400 µM ramucirumab administrated singularly or in combination. AFP (**A**) and DCP (**B**) levels in the cell culture medium were measured after each drug treatment and the respective values divided by the number of viable cells are reported in the graphs after treatment of 48 h. The results of three independent experiments are expressed as relative mean values referred to the control ± SD. *** *p* < 0.0001.

**Figure 6 cancers-11-00787-f006:**
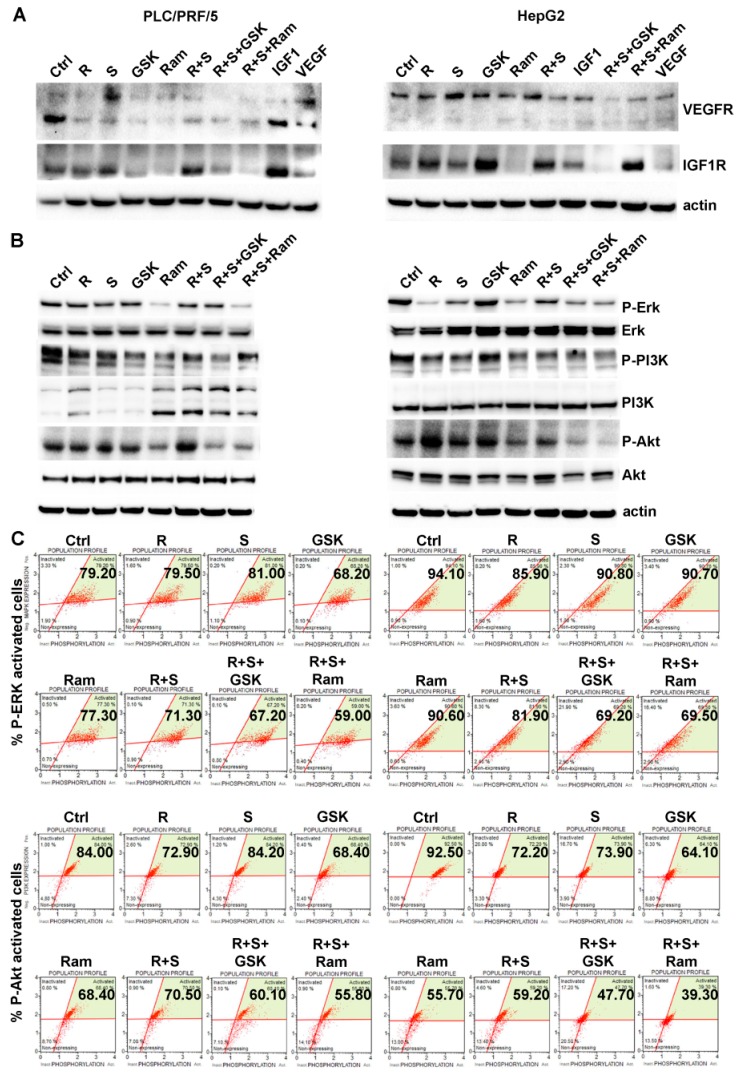
GSK1838705A and ramucirumab potentiate the inhibitory effects of sorafenib/regorafenib combination on the modulation of MAPK and PI3K/Akt signaling in HCC cells. Cells were cultured with 2.5 µM (PLC/PRF/5) or 1 µM (HepG2) sorafenib, 1 µM (PLC/PRF/5) or 0.1 µM (HepG2) regorafenib, 3 µM GSK1838705A and 400 µM ramucirumab administrated singularly or in combination. (**A**) In addition to the described treatments, IGF1 75 ng/mL and VEGF 20 ng/mL recombinant molecules were used as single treatment. Western blotting showing the expression levels of VEGFR and IGF1R after 48 h of single or combined treatments. (**B**) Western blotting showing the expression levels of phosphorylated Erk, PI3K and Akt relative to the total protein expression after 48h of single or combined treatments. (**C**) The Muse Activation Kits were used to evaluate the ERK and Akt phosphorylation relative to the total protein expression after 15 min and 24 h respectively. An example of ERK and Akt activation status after each drug treatment conditions are shown in the relative panels.

**Table 1 cancers-11-00787-t001:** MTT assay relative to each single or combined treatment in PLC/PRF/5, and HepG2 cells. MTT assay performed in cells cultured with 2.5 µM (PLC/PRF/5) or 1 µM (HepG2) sorafenib (S), 1 µM (PLC/PRF/5) or 0.1 µM (HepG2) regorafenib (R), 3 µM GSK1838705A (GSK) and 400 µM ramucirumab (Ram) alone or in combination. The average value of number of colonies derived from three independent experiments, the *p* value and the percentage decreases for the interest group are reported.

PLC/PRF/5 Samples (OD MTT Assay)	“Mann Whitney Test” *p* Value	Decrease (%)
Ctrl vs. R 1.63 vs. 1.34	*** 0.0001	17.8
Ctrl vs. S 1.63 vs. 1.41	*** 0.0001	13.5
Ctrl vs. GSK 1.63 vs. 1.46	*** 0.0001	10.4
Ctrl vs. Ram 1.63 vs. 1.3	*** 0.0001	20.2
R vs. R+S 1.35 vs. 0.95	*** 0.0001	29.6
R vs. R+S+GSK 1.35 vs. 0.72	*** 0.0001	46.7
R vs. R+S+Ram 1.35 vs. 0.46	*** 0.0001	65.9
S vs. R+S 1.41 vs. 0.95	*** 0.0001	32.6
S vs. R+S+GSK 1.41 vs. 0.72	*** 0.0001	48.9
S vs. R+S+Ram 1.41 vs. 0.46	*** 0.0001	67.4
GSK vs. R+S+GSK 1.46 vs. 0.72	*** 0.0001	49.3
Ram vs. R+S+Ram 1.3 vs. 0.46	*** 0.0001	64.6
R+S vs. R+S+GSK 0.95 vs. 0.72	*** 0.0001	24.2
R+S vs. R+S+Ram 0.95 vs. 0.46	*** 0.0001	51.6
**HepG2 Samples** **(OD MTT Assay)**	**“Mann Whitney Test”** ***p*** **Value**	**Decrease** **(%)**
Ctrl vs. R 1.01 vs. 0.88	*** 0.0001	12.9
Ctrl vs. S 1.01 vs. 0.73	*** 0.0001	27.7
Ctrl vs. GSK 1.01 vs. 1.09	ns	+10
Ctrl vs. Ram 1.01 vs. 0.91	ns	10
R vs. R+S 0.88 vs. 0.65	*** 0.0001	26
R vs. R+S+GSK 0.88 vs. 0.41	*** 0.0001	53.4
R vs. R+S+Ram 0.88 vs. 0.54	*** 0.0001	38.6
S vs. R+S 0.74 vs. 0.65	*** 0.0001	12.2
S vs. R+S+GSK 0.74 vs. 0.41	*** 0.0001	44.6
S vs. R+S+Ram 0.74 vs. 0.54	*** 0.0001	27
GSK vs. R+S+GSK 1.09 vs. 0.41	*** 0.0001	62.4
Ram vs. R+S+Ram 0.91 vs. 0.54	*** 0.0001	40.7
R+S vs. R+S+GSK 0.65 vs. 0.41	*** 0.0001	36.9
R+S vs. R+S+Ram 0.65 vs. 0.54	*** 0.0001	16.9

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
