# Peer review of "Ramucirumab and GSK1838705A Enhance the Inhibitory Effects of Low Concentration Sorafenib and Regorafenib Combination on HCC Cell Growth and Motility"

_cancers, 2019, doi:10.3390/cancers11060787_

Round 1

Reviewer 1 Report

Article has an interesting scientific aspect, and possibly in the future of clinical significance. After the extension of laboratory tests, the determination of biochemical markers, i.e. IGF1R and VEGFR2, can be used as tumor markers in patients suffering from hepatocellular cancer.
The authors should correct Figure 2 because it is illegible in its present form.

Author Response

Comments and Suggestions for Authors

Article has an interesting scientific aspect, and possibly in the future of clinical significance. After the extension of laboratory tests, the determination of biochemical markers, i.e. IGF1R and VEGFR2, can be used as tumor markers in patients suffering from hepatocellular cancer.

1- The authors should correct Figure 2 because it is illegible in its present form.

According to the reviewer suggestions we provided to replace the immunofluorescence Ki67 images with the relative graphs in which the fluorescence intensity values were plotted. The results obtained are thus clearer and more immediate. Furthermore, we added graphs relative to cell cycle progression after reported treatments and a representative set of panels showing cell distribution in the G0/G1, S and G2/M phases was displayed in Figure S1.

Reviewer 2 Report

The Authors of the article tested whether Regorafenib/Sorafenib combination may affect the proliferative status of HCC. They showed that simultaneous inhibition of multiple signalling cascades can be approach for HCC management. The article is characterized by novelty and may be interesting for the readers of Cancer.

The paper is good, but the style of the manuscript and the presentation of the results decreases its value.

1.      In my opinion the abstract should be rewritten. The Authors are more focused on what technique they used rather than results that they achieved.

2.      The introduction section could be improved – for instance Authors could explain the differences in the models that they used. Why they decided to use both PLC/PRF/5 and HepG2 cells.

3.      Authors claim that they observed depolymerization of actin cytoskeleton. However, the Figure 4 has poor quality (low magnification and resolution) thus it is difficult to agree with Authors. Further Authors used actin as a reference in western blot analysis and show bands with comparable intensity.

4.      Authors, described anti-proliferative activity of drug combinations based on the Ki-67 staining. In my opinion, this results should be also supported by the analysis of cell distribution in the cell cycle. Can you improve this data?

The strongest part of the paper is well-written discussion section.

Additional remarks are indicated in the manuscript (pdf file).

Author Response

Comments and Suggestions for Authors

The Authors of the article tested whether Regorafenib/Sorafenib combination may affect the proliferative status of HCC. They showed that simultaneous inhibition of multiple signalling cascades can be approach for HCC management. The article is characterized by novelty and may be interesting for the readers of Cancer.

The paper is good, but the style of the manuscript and the presentation of the results decreases its value.

1- In my opinion the abstract should be rewritten. The Authors are more focused on what technique they used rather than results that they achieved.

- The abstract was rewritten giving greater emphasis to the results obtained.

2- The introduction section could be improved – for instance Authors could explain the differences in the models that they used. Why they decided to use both PLC/PRF/5 and HepG2 cells.

- As rightly pointed out by the reviewer, in the revised introduction we specified that the reasons for the choice of the two HCC cell lines are to be attributed to the different content of AFP and DCP of PLC/PRF/5 and HepG2.

3- Authors claim that they observed depolymerization of actin cytoskeleton. However, the Figure 4 has poor quality (low magnification and resolution) thus it is difficult to agree with Authors. Further Authors used actin as a reference in western blot analysis and show bands with comparable intensity.

-The reviewer's observation gives us the opportunity to better clarify an important aspect of our study. In immunofluorescence experiments we evaluated the reduction of the F-actin. Although the small size of each panel (due to the high number of treatments) the reduction of the fluorescent signal of the filamentous form of actin (F-actin) is evident especially after the combined treatments. The polygonal shape of the PLC/PRF/5 cells also makes clearly visible the change in the distribution of fibers that become more globular and placed around the nucleus. Therefore, in the additional material we have included a table (Table S5: DyLight 554 Phalloidin immunofluorescent staining) with the average fluorescence values obtained from each treatment that help in the analysis of the results.

The total content of actin was unchanged and can therefore be used as house-keeping protein in Western Blotting experiments.

4- Authors, described anti-proliferative activity of drug combinations based on the Ki-67 staining. In my opinion, this results should be also supported by the analysis of cell distribution in the cell cycle. Can you improve this data?

-Given the importance of the effects of the treatments indicated in cellular proliferation, we accepted the suggestion of the reviewer and introduced the analysis of progression in the cell cycle. Hoping that the analysis of cell proliferation is now improved, we presented the relative results in the revised manuscript, in new Figure 2 and in Figure 1S.

The strongest part of the paper is well-written discussion section.

Additional remarks are indicated in the manuscript (pdf file).

-We have rewritten all the highlights. In the revised manuscript, all the revisions are clearly highlighted in red.

Reviewer 3 Report

This study evaluated effects of inhibitors of IGF1R and VEGR on the low dose combo of sorafenib and regorafenib.

Minor revisions

1. It is unclear whether one can use one agent of a class and claims the efficacy as a class. It would be more appropriate to claim that if two agents of a class were used. In this case, it would be more appropriate to mention the name of the agent ramucirumab and GSK—in the title, if authors do not wish to add more studies

2. Lenvatinib should be mentioned in the introduction instead of in the discussion.

Author Response

Comments and Suggestions for Authors

This study evaluated effects of inhibitors of IGF1R and VEGR on the low dose combo of sorafenib and regorafenib.

Minor revisions

1. It is unclear whether one can use one agent of a class and claims the efficacy as a class. It would be more appropriate to claim that if two agents of a class were used. In this case, it would be more appropriate to mention the name of the agent ramucirumab and GSK—in the title, if authors do not wish to add more studies

-We thank the reviewer for the suggestion that allows us to be more precise. We have changed the title replacing the names of the two compounds used to those of the classes to which they belong.

2. Lenvatinib should be mentioned in the introduction instead of in the discussion.

-According to the reviewer we mentioned lenvatinib in the revised introduction.